# Cannabidiol Interacts Antagonistically with Cisplatin and Additively with Mitoxantrone in Various Melanoma Cell Lines—An Isobolographic Analysis

**DOI:** 10.3390/ijms23126752

**Published:** 2022-06-17

**Authors:** Paweł Marzęda, Paula Wróblewska-Łuczka, Małgorzata Drozd, Magdalena Florek-Łuszczki, Katarzyna Załuska-Ogryzek, Jarogniew J. Łuszczki

**Affiliations:** 1Department of Pathophysiology, Medical University of Lublin, 20-090 Lublin, Poland; pawel.marzeda@umlub.pl (P.M.); paula.wroblewska-luczka@umlub.pl (P.W.-Ł.); drozd.malg@gmail.com (M.D.); katarzyna.zaluska-ogryzek@umlub.pl (K.Z.-O.); 2Department of Medical Anthropology, Institute of Rural Health, 20-950 Lublin, Poland; florek.magdalena@imw.lublin.pl

**Keywords:** cannabinoids, cannabidiol, melanoma, in vitro study, drug interactions

## Abstract

The medical application of cannabidiol (CBD) has been gathering increasing attention in recent years. This non-psychotropic cannabis-derived compound possesses antiepileptic, antipsychotic, anti-inflammatory and anxiolytic properties. Recent studies report that it also exerts antineoplastic effects in multiple types of cancers, including melanoma. In this in vitro study we tried to reveal the anticancer properties of CBD in malignant melanoma cell lines (SK-MEL 28, A375, FM55P and FM55M2) administered alone, as well as in combination with mitoxantrone (MTX) or cisplatin (CDDP). The effects of CBD on the viability of melanoma cells were measured by the MTT assay; cytotoxicity was determined in the LDH test and proliferation in the BrdU test. Moreover, the safety of CBD was tested in human keratinocytes (HaCaT) in LDH and MTT tests. Results indicate that CBD reduces the viability and proliferation of melanoma-malignant cells and exerts additive interactions with MTX. Unfortunately, CBD produced antagonistic interaction when combined with CDDP. CBD does not cause significant cytotoxicity in HaCaT cell line. In conclusion, CBD may be considered as a part of melanoma multi-drug therapy when combined with MTX. A special attention should be paid to the combination of CBD with CDDP due to the antagonistic interaction observed in the studied malignant melanoma cell lines.

## 1. Introduction

Melanoma is a malignant tumor of pigment-producing cells, usually localized in the skin. The incidence rates of this cancer remain high and still increase, mostly in elderly patients [1,2]. Although mortality rates for melanoma have dropped in recent years, mostly due to prevention and development of novel therapies, prognosis when distant metastases occur is considered to be poor [2]. The cost of the novel melanoma therapies, including Ipilimumab and Nivolumab, remains high [3]. Targeted therapies improve outcomes, but for many patients it is not uncommon to present only a partial response and relapse during follow-up. Moreover, there is concern of the high toxicity of these agents [4]. Therefore, there is still a need for cheap and effective treatment alternatives in patients with melanoma. Despite various currently available drugs, therapeutic regimens and methods in the treatment of malignant melanoma, the choice of the most effective treatment option is still a challenging issue for clinicians [5,6,7].

Cannabidiol (CBD) is one of the 120 known phytocannabinoids, i.e., compounds extracted from the plant *Cannabis sativa*, which interact with the mammalian endocannabinoid system [8]. In recent years, medical applications of CBD gathered great attention, not only scientists and clinicians. The crucial feature of the CBD, marking its potential usefulness is lack of psychoactive effects, in contrast to other cannabinoids (i.e., Δ9-tetrahydrocannabinol [THC]). For instance, THC demonstrates anxiolytic, antiepileptic, antipsychotic effect, reduces inflammation and acts as a neuroprotective agent [9,10]. Numerous reports of beneficial effects of CBD in the setting of various neoplasms have been published earlier [11,12].

CBD has been successfully implemented in patients with various cancers, usually in metastatic stages, e.g., breast, prostate, oesophageal, with spectacular anti-tumor effects in patients with glioma. However, clinical evidence is limited to case reports [4,13]. Preclinical studies provide positive results in colorectal/colon cancer, prostate cancer, lung cancer, brain cancer, endothelial cells cancer and melanoma [14,15]. Antineoplastic properties of CBD in various cancers include the induction of apoptosis, autophagy, cell cycle arrest, necrosis, the modulation of tumor microenvironment and reduction in angiogenesis, invasiveness and inflammation [12,14,15]. CBD alone or in combination with THC can be applied as a supplementary treatment of persistent symptoms associated with malignant disease or side effects of cytostatic therapy, such as nausea and vomiting, chronic pain, loss of appetite, cachexia, sleep problems, anxiety or even depression [16,17,18,19]. In contrast to THC, CBD does not reveal psychotropic properties, responsible for legal controversy around marijuana’s accessibility [4]. Accumulating experimental evidence indicates that CBD inhibits cellular uptake of anandamide (an endogenous CB1 receptor agonist) and has anti-inflammatory and immunosuppressive properties [20], and thus, CBD is efficacious in some skin disorders, including eczema, psoriasis, pruritus and inflammation [21]. CBD can be used in the regulation of pain in cancer patients [22]. Cannabinoid agonists modulate various cancer-related pathways that can lead to the blockade of the cell cycle, inhibition of the cell proliferation and cell death [23]. Cannabinoid ligands (including CBD) possess the anti-invasiveness and anti-metastatic actions [24,25,26]. Cannabinoids also exert the anti-angiogenesis effects by blocking the activation of the vascular endothelial growth factor pathway [27]. Molecular studies have revealed that cannabinoids can inhibit the PI3K-Akt and activate MAPK pathways, resulting in apoptotic cell death, respectively [28]. Additionally, cannabinoids can inhibit the AKT/mTORC1 axis, and thus induce cell death by autophagy [29]. CBD is an agonist of PPAR-γ, TRPV1 and TPRV2 [30]. CBD inhibits the expression of the anti-apoptotic proteins AKT (B) and increases the release of cytochrome c from mitochondria, and thus enhancing apoptosis [31]. Moreover, CBD-enriched hemp oil promotes the proliferation of normal fibroblasts and promotes apoptosis in osteosarcoma cells through p53-independent mechanisms [32]. CBD also inhibits B16 mouse melanoma cells in in vitro study [33].

Taking into account the facts that CBD can be effective in the treatment of various cancers and is efficacious in some skin disorders, we attempted to establish if CBD reduces the viability of various human melanoma cell lines (SK-MEL 28, A375, FM55P and FM55M2) and if CBD can be considered as a part of multi-drug chemotherapy when combined with two chemotherapeutic drugs commonly used in experimental in vitro studies, i.e., cisplatin (CDDP) and mitoxantrone (MTX).

Cisplatin (CDDP) is one of the most potent antitumor agents whose cytotoxic activity is mediated by its interaction with DNA to form DNA adducts, which activate several signal transduction pathways (including ATR, p53, p73 and MAPK), leading finally to cell apoptosis [34,35,36,37,38,39]. Mitoxantrone (MTX—an anthracenedione antineoplastic agent) is a type II topoisomerase inhibitor which disrupts DNA synthesis and DNA repair in cancer cells by intercalation between DNA bases. MTX causes DNA aggregation and compaction and delays cell cycle progression in late S phase [40,41,42,43,44].

Due to the various molecular mechanisms of the action of the studied drugs (CBD, MTX and CDDP), it is expected that the two-drug combinations of CBD with CDDP or CBD with MTX should exert synergistic interactions in terms of the anti-proliferative effects of these drugs in various human melanoma cell lines (SK-MEL 28, A375, FM55P and FM55M2). The assessment of the interaction between CBD and CDDP or CBD and MTX was performed by means of the isobolographic analysis, which is considered to be a gold standard in evaluation of types of drug–drug interaction in cancer studies [45,46,47].

## 2. Results

Cannabidiol (CBD), cisplatin (CDDP) and mitoxantrone (MTX) reduced the viability of various human melanoma cell lines (i.e., SK-MEL 28, A375, FM55P and FM55M2) in a concentration-dependent manner, when applied separately (Figure 1, Table 1). Moreover, CBD reduced the viability of the normal human keratinocytes (HaCaT) but in high concentrations. It is noteworthy that none of the solvents used in the respective control groups, not ethanol, phosphate buffered saline (PBS) nor dimethyl sulfoxide (DMSO), tested in relevant concentrations affected the viability of melanoma cells (data not shown). The experimentally derived median inhibitory concentration (IC_50_) values for CBD, CDDP and MTX in various melanoma cell lines are presented in Table 1.

The following step was the quantification of the cytotoxicity of CBD in malignant melanoma cell lines and normal keratinocytes in the LDH test. The diagrams show that the cytotoxicity of CBD in various malignant melanoma cell lines grows in a concentration-dependent manner, while CBD has no significant impact on normal human keratinocytes (Figure 2).

Next, the BrdU test was performed to determine the effect of CBD on the proliferation of malignant melanoma cells (A375, SK-MEL 28, FM55P and FM55M2) and normal human keratinocytes (HaCaT). As presented below, CBD inhibited the proliferation of all tested melanoma cell lines, where the effect was the strongest in the cell line obtained from a primary tumor—the FM55P cell line (Figure 3). CBD in the highest tested concentration of 10 µg/mL inhibits the proliferation of normal human keratinocytes (HaCaT) in approx. 50% (Figure 3).

The next step was determining the anti-proliferative effects of CBD administered alone and in combination with CDDP or MTX to the A375, SK-MEL 28, FM55P and FM55M2 melanoma cell lines. All mentioned cell lines were incubated with different concentrations of CBD and CDDP or MTX, based on the established IC_50_ values. The obtained results presented the concentration-dependent reduction in cancer cell viability (Figure 4 and Figure 5).

With isobolography, the combination of CBD and CDDP at the fixed ratio of 1:1 exerted antagonistic interaction in the SK-MEL 28 cell line (Figure 6A). The statistical analysis of the data performed by means of the Student’s *t*-test with Welch correction revealed that the IC_50 mix_ value significantly differed (*p* < 0.05) from the IC_50 add_ values (Table 2; *t* = 2.013; df = 196.1; *p* = 0.045; Figure 6A). The similar antagonistic interaction was observed for the mixture of CBD with CDDP at the fixed ratio of 1:1 in the A375 cell line (Figure 6B). Statistical analysis by using the Student’s *t*-test with Welch correction confirmed that the IC_50 mix_ value significantly differed (*p* < 0.01) from the IC_50 add_ values (Table 2; *t* = 2.728; df = 141.4; *p* = 0.0072; Figure 6B). The combination of CBD with CDDP at the fixed ratio of 1:1 exerted antagonistic interaction in the FM55P cell line (Figure 6C). Statistical analysis of data with Student’s *t*-test with Welch correction revealed that the IC_50 mix_ value significantly differed (*p* < 0.05) from the IC_50 add_ values (Table 2; *t* = 2.063; df = 210; *p* = 0.040; Figure 6C). In contrast, the combination of CBD with CDDP at the fixed ratio of 1:1 produced additive interaction in the FM55M2 melanoma cells (Figure 6D). Statistical analysis by using the Student’s t-test with Welch correction revealed that the IC_50 mix_ value did not significantly differ from the IC_50 add_ values (Table 2; *t* = 0.310; df = 207.9; *p* = 0.757; Figure 6D).

The isobolographic analysis of the interaction between CBD and MTX revealed that the two-drug combination exerted additive interaction in the SK-MEL 28 melanoma cells (Figure 7A). The statistical analysis of the data with Student’s *t*-test with Welch correction confirmed that the IC_50 mix_ value did not differ from the IC_50 add_ values (Table 3; *t* = 1.617; df = 211.9; *p* = 0.1073; Figure 7A). The additive interaction was observed for the mixture of CBD with MTX in the A375 melanoma cells (Figure 7B). The statistical analysis of data by using the Student’s *t*-test with Welch correction revealed that the IC_50 mix_ value did not differ from the IC_50 add_ values (Table 3; *t* = 0.4379; df = 210; *p* = 0.6619; Figure 7B). The same situation was observed for the combination of CBD with MTX that produced additive interaction in the FM55M2 melanoma cells (Figure 7C). The statistical analysis of the data performed by means of the Student’s *t*-test with Welch correction revealed that the IC_50 mix_ value did not differ from the IC_50 add_ values (Table 3; *t* = 1.261; df = 248.9; *p* = 0.2085; Figure 7C). The combination of CBD with MTX exerted additive interaction in the FM55P melanoma cells because no significant difference was observed with Student’s t-test with Welch correction between the IC_50 mix_ and IC_50 add_ values (Table 3; *t* = 1.365; df = 210; *p* = 0.1738; Figure 7D).

## 3. Discussion

Melanoma is considered to bear one of the highest mutation frequencies, and, therefore, has vast tumor heterogeneity [48,49]. This impedes the response to the immune therapies and patients’ survival [6,7,49]. Bearing that in mind we investigated the effectiveness of CBD in different tumors we tested in various cell lines.

Our study shows that CBD reduces melanoma cells viability in concentration-dependent manner. The most vulnerable cell line to the effects of CBD was the melanoma cell line obtained from primary tumor—FM55P (IC_50_ = 3.81 ± 0.35 µg/mL), and the most resistant was SK-MEL 28 cell line (IC_50_ = 7.75 ± 0.29 µg/mL). Similar results were obtained by Burch et al. in B16 mice melanoma cells, however, the range of used CBD concentrations was much higher, up to 200 µg/mL [33]. Simmerman et al. presented that the CBD administered in murine B16F10 cell line reduced the growth of melanoma cells and extended the survival time in animals [50]. The authors also compared the effectiveness of CBD to that of CDDP, since CDDP presented a better survival time and a greater tumor growth reduction, but it worsened the quality of the patient’s life [50]. However, these two compounds were neither co-administered, nor tested in any different cell line. Our results revealed for the first time that CBD and CDDP administered together exerted antagonistic interactions or additivity with a tendency towards antagonism in all the studied cell lines. Quite similar results on the co-administration of CBD and CDDP were obtained by Deng et al. in glioblastoma cell lines [51].

It should be stressed that in this study we determined the anti-proliferative effects of CBD and CDDP and MTX in two closely related cell lines—primary-derived melanoma (FM55P) and metastatic-derived melanoma (FM55M2). The difference between these two cell lines was reported earlier, documenting various (quite different) types of interactions when combined naturally occurring coumarins with CDDP [52]. It seems that diverse cell colonies creating FM55M2 cell line were more sensitive to the applied mixture of coumarins with CDDP, and thus produced synergistic or additive interactions in the MTT assay [52]. In this study, we also observed such diversity in the anti-proliferative activity of the two-drug mixture in these two cell lines. In the FM55M2 cell line, the additivity was observed while the same combination exerted antagonistic interaction in the FM55P cell line. It seems that metastatic-derived cell line was more sensitive to the applied therapy than its primary counterpart. Different colony cells might be responsible for such effects in the in vitro study. Although this is a hypothesis, it can explain the observed difference in the effects in this study.

On the other hand, the combination of CBD and MTX produces additive interactions, with the most potent effect in FM55P cell line. These results stand in line with data obtained from the study performed on canine urothelial carcinoma cells, where CBD presented synergistic effects with MTX [53]. The potential mechanism of the action responsible for this interaction might be inhibition of ABCG2/Abcg2 multidrug transporter, which is responsible for the transportation of MTX out of the cancer cells [54]. CBD might slow down the function and/or activity of this multi-drug transporter contributing to the elevation of MTX concentrations in the melanoma cells, and thus producing favorable effects.

In LDH test we found that CBD reduces HaCaT cells viability in concentrations higher than those needed to significantly reduce the viability of melanoma cell lines. Other authors presented that CBD did not affect the viability of HaCaT cells in concentrations up to 20 μM (6.29 µg/mL) during exposition up to 24 h [55]. Our exposition time was longer (72h) and the applied concentrations of CBD were higher (up to 10 µg/mL).

The majority of currently available drugs contain CBD combined with THC in various dose ratios. The study of Armstrong et al. investigated in vitro the effectiveness of CBD administered together with THC in equal ratio, in doses of 0.5 μM up to 2.5 μM in SK-MEL 28, A375 and CHL-1 cell lines [56]. Co-administration of these two cannabinoids lead to greater reduction in melanoma cell viability than THC administered alone [56]. Similar results were obtained in the in vivo study on the CHL-1 mice line BRAF wild-type melanoma xenografts, where the addition of CBD to THC enhanced the inhibition of tumor growth [56]. This indicates that CBD and THC exert their actions via different molecular pathways [57].

CBD can potentially be used in phytoradiotherapy, where it enhances the therapeutic efficacy of radiotherapy and mitigate its side effects [3,58]. However, it was not yet tested in the setting of melanoma. The radiation in melanoma has limited, but potential application [59].

Isobolographic analysis is the method of choice allowing precisely classifying the type of interactions observed experimentally between two or more drugs used in mixture [45,60,61,62]. In the treatment of neoplasms and cancers, the therapeutic use of one drug is very often insufficient and two or more drugs (chemotherapeutics) applied simultaneously are needed to efficiently eliminate reproducing cancer cells. Thus, polytherapy is frequently used and preferred by clinicians during chemotherapeutic “combat” against cancer cells. Chemotherapy must be adjusted to the patient’s condition and the type of the neoplasm, in order not to harm the patient’s life. Isobolographic analysis can readily provide information whether the chosen drug combination is effective or not. From a pharmacological viewpoint, one can distinguish the three main types of interactions occurring between the anti-cancer drugs, including, supra-additivity (synergy), additivity and sub-additivity (antagonism) [45,52,60,61,62]. Synergy is observed if two or more anti-cancer drugs cooperate and mutually potentiate their anti-cancer effects in terms of killing the reproducing cancer cells. In such a situation, low concentrations of drugs used in a mixture can eliminate many more cancer cells than could be expected if the drugs were used separately. In other words, if one drug inhibits 10% of the reproducing cancer cells and the second drug also 10% of the cancer cells, then synergy occurs if cancer cells are eliminated by the drug mixture in 60%. Of note, synergy is the most desirable interaction between anticancer drugs. Additivity is observed if the drugs in the mixture eliminate the cancer cells as a result of the summation of the effects produced by the drugs used separately. In other words, if one drug inhibits 10% and the second drug also 10% of the cancer cells, then additivity occurs if cancer cells will be eliminated by the drug mixture in 20%. Antagonism is observed if the drugs in the mixture produce lower effects than would be expected after the separate application of the drugs. In the case of sub-additivity, one drug used separately inhibits 50% of the cancer cells and the second drug also 50% when used alone but their combination only eliminates 75% of the cancer cells. The isobolographic analysis of the interaction is a research method joining both the mathematical calculation of IC_50 add_ values with statistical comparison between theoretically calculated IC_50 add_ values (accepted as additive) with experimentally derived IC_50 mix_ values, respectively. Details concerning the isobolographic analysis have been published elsewhere [63,64,65,66].

## 4. Materials and Methods

### 4.1. Cell Lines

Primary (FM55P) and metastatic (FM55M2) malignant melanoma cells were purchased from European Collection of Cell Cultures (ECACC) and cultured in RPMI—1640 Medium (Sigma-Aldrich, St. Louis, MA, USA). Another two cell lines A375 (primary malignant melanoma) and SK-MEL 28 (metastatic malignant melanoma) were purchased from the American Type Culture Collection (ATCC) and cultured in Dulbecco’s Modified Eagle’s Medium—high glucose (DMEM) (Sigma-Aldrich) and Eagle’s minimal essential medium (EMEM), respectively. All culture media were supplemented with 10% Fetal Bovine Serum (FBS; Sigma-Aldrich) and 1% of penicillin/streptomycin (Sigma-Aldrich). Cultures were kept at 37 °C in a humidified atmosphere of 95% air and 5% CO_2_. The cells were grown to 80% confluence.

### 4.2. Drugs

Cisplatin (CDDP—Sigma-Aldrich) was dissolved in phosphate buffered saline (PBS) with Ca^2+^ and Mg^2+^. Mitoxantrone (MTX—Sigma-Aldrich) was dissolved in DMSO as stock solutions. The cannabidiol (CBD—Tocris, Bristol, UK) was dissolved in ethanol as stock solutions in concentration of 5 mg/mL. The drugs were dissolved to the respective concentrations with culture medium before use.

### 4.3. Cell Viability Assessment

SK-MEL 28, A375, FM55P and FM55M2 cells were placed on 96-well plates (Nunc, Roskilde, Denmark) at a density of 2 × 10^4^ cells/mL, 3 × 10^4^ cells/mL, 2 × 10^4^ cells/mL and 2 × 10^4^ cells/mL, respectively. Next day, the culture medium was removed and cells were exposed to serial dilutions of CBD, CDDP and MTX in fresh culture medium. Cell viability was assessed after 72 h by means of MTT test, in which the yellow tetrazolium salt (MTT) is metabolized by viable cells to purple formazan crystals. Cells were incubated for 3 h in the MTT solution (5 mg/mL, Sigma-Aldrich). Formazan crystals were solubilized overnight in sodium dodecyl sulfate (SDS) buffer (10% SDS in 0.01 N HCl) and the product was determined spectrophotometrically by measuring absorbance at 570 nm wavelength using microplate spectrophotometer (Ledetect 96, Labexim Products, Lengau, Austria). Each treatment was performed in triplicate and each experiment was repeated 3 times.

### 4.4. Cell Proliferation Assay

Cell Proliferation Elisa, BrdU Kit (Roche Diagnostics, Mannheim, Germany) was used following manufacturer’s instructions. Optimized amounts of A375 (2 × 10^4^/mL) SK-MEL 28 (3 × 10^4^/mL), FM55P (2 × 10^4^/mL), FM55M2 (2 × 10^4^/mL) cells were placed on a 96-well plate (Nunc) (100 μL/well). On the next day, the cancer cells were treated with increased concentrations of CBD for 48 h, followed by 10 µL/well BrdU Labeling Solution (100 µM) was added and cells were reincubated for additional 24 h at 37 °C. Then, the culture medium was removed and cells were fixed in FixDenat solution (200 µL/well) (30 min., room temperature—RT). The working solution of anti-BrdU antibody coupled with horseradish peroxidase (anti-BrdU-POD) were subsequently added (100 µL/well) (90 min, RT) and detected using tetramethylobenzidine substrate (TMB) (100 µL/well) (30 min, RT). A total of 1 M sulfuric acid was added (25 µL/well) to stop enzymatic reaction, and quantitation was performed spectrophotometrically at 450 nm using microplate spectrophotometer (Ledetect 96, Labexim Products).

### 4.5. Cytotoxicity Assessment—LDH Assay

Optimized amounts of A375 (2 × 10^4^/mL) SK-MEL 28 (3 × 10^4^/mL), FM55P (2 × 10^4^/mL), FM55M2 (2 × 10^4^/mL) and normal human keratinocytes HaCaT (1 × 10^4^/mL) cells were placed on 96-well plates (Nunc). Next day, cells were washed in PBS, and then exposed to increasing concentrations of CBD in fresh culture medium. The cytotoxicity was estimated based on the measurement of cytoplasmic lactate dehydrogenase (LDH) activity released from damaged cells after 72 h exposure to CBD. LDH assay was performed according to manufacturer’s instruction (Cytotoxicity Detection KitPLUS LDH) (Roche). Briefly, 50 µL of cell medium was collected from each well, then 50 µL of reaction mixture (freshly prepared) was added and incubated for 30 min at RT. Finally, 25 µL of Stop solution was added to each well on the 96-well plate. Absorbance was measured at two different wavelengths, one being the “measurement wavelength” (492 nm) and the other “reference wavelength” (690 nm) using microplate spectrophotometer (Ledetect 96, Labexim Products). Maximum LDH release (positive control) was achieved by the addition of Lysis buffer to untreated control cells. The average values of the culture medium background were subtracted from all values of experimental wells and the percentage of death cells was calculated in relation to the maximum LDH release.

### 4.6. Isobolographic Analysis of Interactions

Log-probit analysis was used to determine the percentage of inhibition of cell viability per concentration of CBD, CDDP and MTX when administered singly in the A375, SK-MEL 28, FM55P and FM55M2 cell lines measured in vitro by the MTT assay. Subsequently, from the log-probit concentration–response lines, median inhibitory concentrations (IC_50_ values) of CBD and CDDP, MTX were calculated [67]. The log-probit method is accepted as a method of choice when determining the IC_50_ values, as recommended earlier [68]. Additionally, the probit-type concentration–response curves for the studied drugs (CBD and CDDP, CBD and MTX) were verified in terms of their mutual parallelism based on the linear Loewe additivity model [66,67,68,69,70]. In such a situation, the drug concentrations were transformed to logarithms and their antiproliferative effects were transformed to probits. The log-probit analysis revealed that CBD had its concentration–response line non-parallel to CDDP and MTX in all the tested cell lines (Figure 4 and Figure 5). Of note, lack of parallelism between the analyzed lines forced us to perform type I isobolographic analysis for non-parallel concentration–response effect lines with additivity defined graphically as an area bounded by two lower and upper isoboles of additivity [63,64,66,68,69]. Thus, the test of parallelism is obligatory in combination experiments to properly classify interaction occurring between drugs [62,63,65,68]. Next, the median additive inhibitory concentrations (IC_50 add_) for the mixture of CBD with CDDP or MTX, which theoretically should inhibit 50% of cell viability, were calculated as demonstrated earlier [66]. The assessment of the experimentally derived IC_50 mix_ at the fixed ratio of 1:1 was based on the concentration of the mixture of CBD and CDDP or MTX that inhibited 50% of cell viability in melanoma cell lines measured in vitro by the MTT assay. In this study, the experimental evaluation of interaction between two drugs in mixture was performed when the drugs were applied in a constant fixed ratio combination of 1:1 [68,70]. In other words, all the studied drugs must exert the similar effect (i.e., inhibits cell proliferation in 50%, which corresponds in approx. to the IC_50_ value) so as to determine the effects exerted by the drugs in mixture. If one of the studied drugs is inactive in this experimental approach, the type II isobolographic analysis must be conducted, which considerably differs from the type I isobolographic analysis [60,61,62].

### 4.7. Statistical Analysis

GraphPad Prism 8.0 Statistic Software was used for statistical analysis. One-way analysis of variance (ANOVA test) for multiple comparisons followed by Tukey’s significance test was used. Data are expressed as the mean ± standard error (SEM) (* *p* < 0.05, ** *p* < 0.01, *** *p* < 0.001, **** *p* < 0.0001). The IC_50_ and IC_50 mix_ values for CBD and CDDP or MTX administered alone or in combination at the fixed ratio of 1:1 were calculated by computer-assisted log-probit analysis [67]. The experimentally derived IC_50 mix_ values for the mixture of CBD with CDDP and CBD with MTX were statistically compared with their respective theoretical additive IC_50 add_ values by the use of unpaired Student’s t-test with Welch correction, as described earlier [70].

## 5. Conclusions

This study confirms that CBD might be a promising agent in the therapy of melanoma when combined with MTX, however, further research is needed to determine the CBD interactions with currently used drugs. Unfortunately, the combination of CBD with CDDP might not be favorable due to the antagonistic interaction in terms of the anti-proliferative effects of both the drugs used in the mixture. More advanced studies are required to elucidate the mechanism(s) responsible for such antagonistic interactions in various human malignant melanoma cell lines.

## Figures and Tables

**Figure 1 ijms-23-06752-f001:**
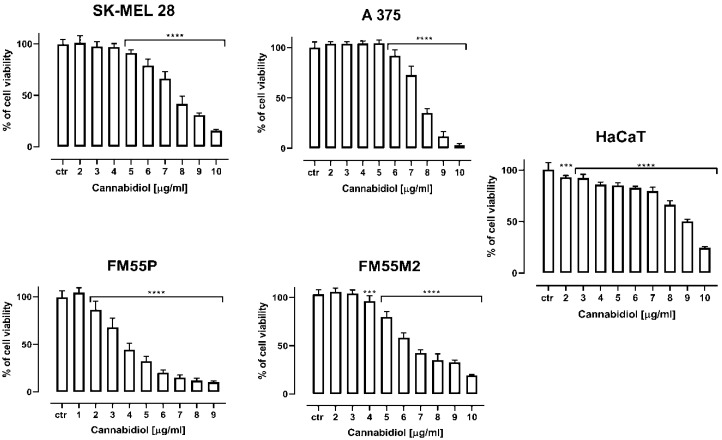
The effect of cannabidiol (CBD) on the viability of malignant melanoma cancer cell lines (SK-MEL 28, A375, FM55P, FM55M2) and the normal human keratinocytes (HaCaT) was measured by means of MTT assay after 72 h. Columns represent mean ± SEM (*** *p* < 0.001 and **** *p* < 0.0001).

**Figure 2 ijms-23-06752-f002:**
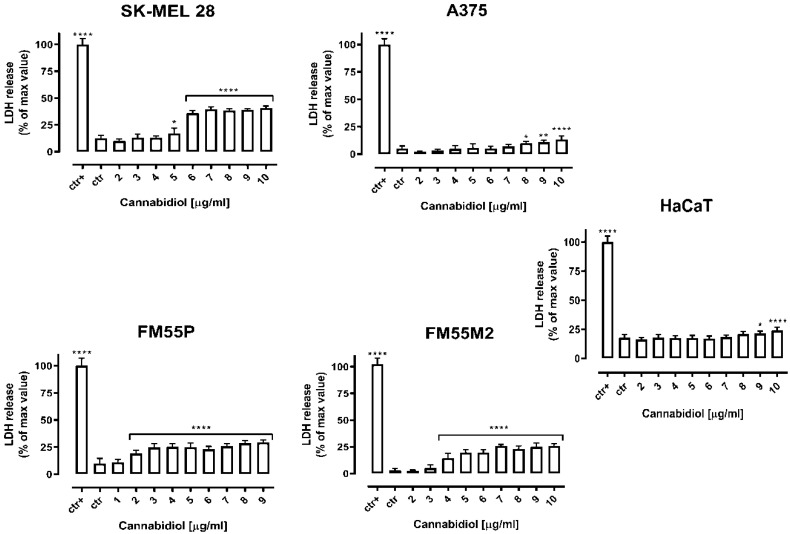
Cytotoxicity of cannabidiol (CBD) to malignant melanoma cells (A375, SK-MEL 28, FM55P and FM55M2) and normal human keratinocytes (HaCaT). Lactate dehydrogenase ELISA kit was used to quantify cytotoxicity by measuring LDH activity released from damaged cells. Normal keratinocytes cells and melanoma cells were incubated for 72 h alone or in the presence of CBD (1–10 µg/mL). The results are presented as the percentage in LDH release to the medium by treated cells versus cells grown in control medium (ctr) and cells treated with Lysis buffer (ctr+). Data are presented as mean ± SEM (* *p* < 0.05, ** *p* < 0.01 and **** *p* < 0.0001).

**Figure 3 ijms-23-06752-f003:**
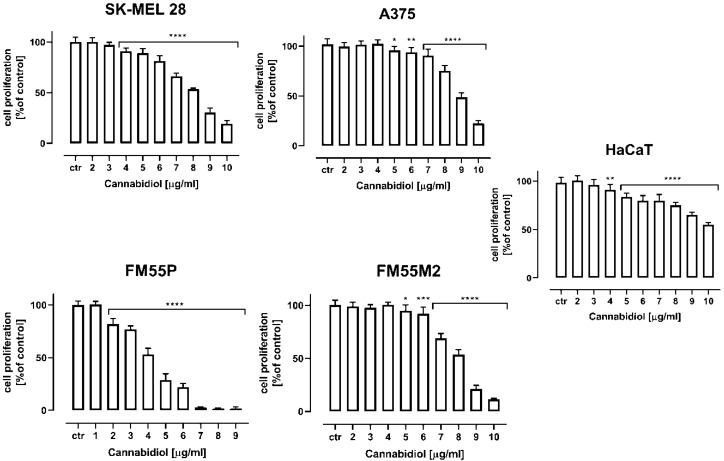
The effect of cannabidiol (CBD) on the proliferation of malignant melanoma cell lines (A375, SK-MEL 28, FM55P and FM55M2) and normal human keratinocytes (HaCaT) measured by means of BrdU assay after 72 h. Results are presented as mean ± SEM (* *p* < 0.05, ** *p* < 0.01; *** *p* < 0.001: **** *p* < 0.0001).

**Figure 4 ijms-23-06752-f004:**
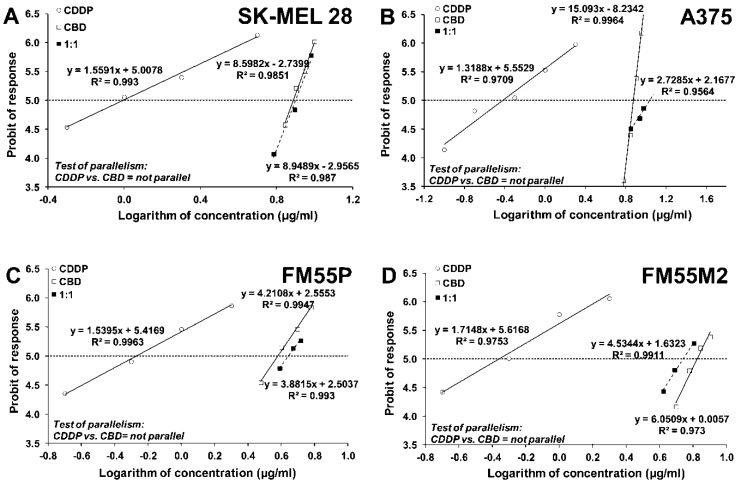
Concentration–effect lines for CBD and CDDP administered alone and in combination in the fixed-ratio of 1:1, illustrating the anti-proliferative effects of the drugs in the malignant melanoma cell lines: SK-MEL 28 (**A**), A375 (**B**), FM55P (**C**) and FM55M2 (**D**) measured in vitro by the MTT assay. Test for parallelism confirmed that the experimentally determined concentration–effect lines for CBD and CDDP (administered alone) are mutually non-parallel to each other in SK-MEL 28, A375, FM55P and FM55M2 cell lines.

**Figure 5 ijms-23-06752-f005:**
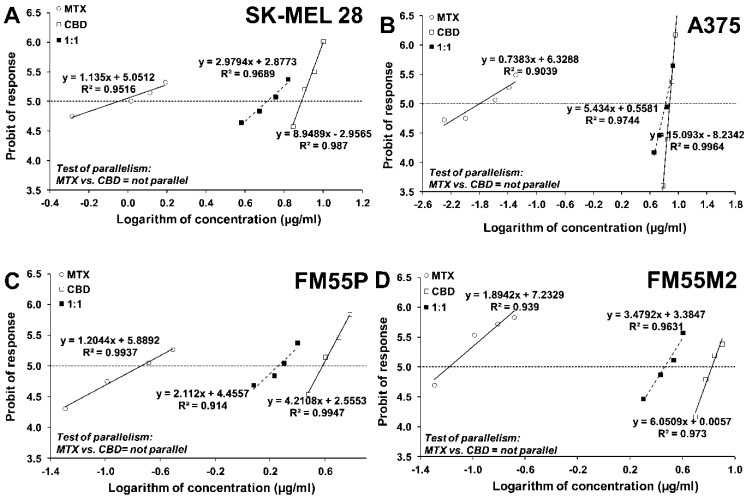
Concentration–effect lines for CBD and MTX administered alone and in combination in the fixed-ratio of 1:1, illustrating the anti-proliferative effects of the drugs in the malignant melanoma cell lines: SK-MEL 28 (**A**), A375 (**B**), FM55P (**C**) and FM55M2 (**D**) measured in vitro by the MTT assay. Test for parallelism revealed that the experimentally determined concentration–response lines for CBD and MTX (administered alone) are not parallel to one another in SK-MEL 28, A375, FM55P and FM55M2 melanoma malignant cell lines.

**Figure 6 ijms-23-06752-f006:**
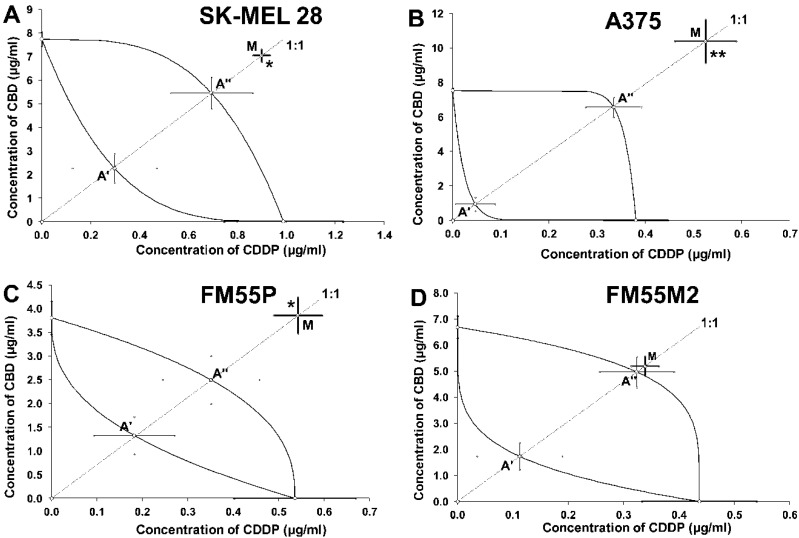
Isobolograms showing interactions between cannabidiol (CBD) and cisplatin (CDDP) with respect to their anti-proliferative effects on SK-MEL 28 (**A**), A375 (**B**), FM55P (**C**) and FM55M2 (**D**) malignant melanoma cell lines measured in vitro by the MTT assay. Points A’ and A” depict the theoretically calculated IC_50 add_ values for both lower and upper isoboles of additivity, respectively. The point M represents the experimentally derived IC_50 mix_ value for total concentration of the mixture of CBD and CDDP that produced a 50% anti-proliferative effect in malignant melanoma cell lines measured in vitro by the MTT assay. * *p* < 0.05 and ** *p* < 0.01 (Student’s *t*-test with Welch correction).

**Figure 7 ijms-23-06752-f007:**
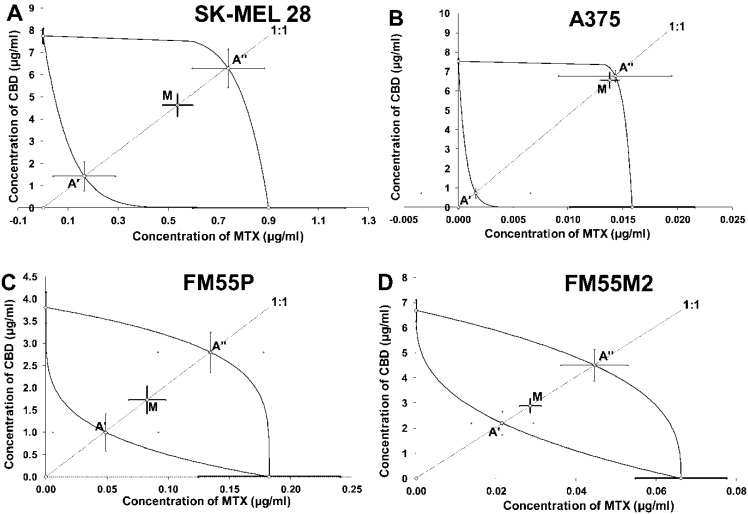
Isobolograms showing interactions between cannabidiol (CBD) and mitoxantrone (MTX) with respect to their anti-proliferative effects on SK-MEL 28 (**A**), A375 (**B**), FM55P (**C**) and FM55M2 (**D**) malignant melanoma cell lines measured in vitro by the MTT assay. Points A’ and A” depict the theoretically calculated IC_50 add_ values for both lower and upper isoboles of additivity, respectively. The point M represents the experimentally derived IC_50 mix_ value for total concentration of the mixture of CBD and MTX that produced a 50% anti-proliferative effect in malignant melanoma cell lines measured in vitro by the MTT assay.

**Table 1 ijms-23-06752-t001:** The anti-proliferative effects of cannabidiol (CBD), cisplatin (CDDP) and mitoxantrone (MTX) administered singly in human malignant melanoma cell lines measured in vitro by the MTT assay.

Drug	SK-MEL 28	A375	FM55P	FM55M2
CBD	7.75 ± 0.29	7.53 ± 0.13	3.81 ± 0.35	6.69 ± 0.42
MTX	0.90 ± 0.30	0.016 ± 0.005	0.18 ± 0.06	0.07 ± 0.01
CDDP	0.99 ± 0.21	0.38 ± 0.07	0.54 ± 0.13	0.44 ± 0.10

Data are median inhibitory concentrations (IC_50_) values in µg/mL (± S.E.M.).

**Table 2 ijms-23-06752-t002:** Isobolographic analysis of interactions between CBD and CDDP (at the fixed ratio of 1:1) in melanoma malignant cell lines.

Cell Line	IC_50 mix_ (μg/mL)	n _mix_	Lower IC_50 add_ (μg/mL)	n _add_	Upper IC_50 add_ (μg/mL)	Interaction
SK-MEL 28	7.95 ± 0.31 *	96	2.56 ± 0.78	158	6.15 ± 0.84	Antagonism
A375	10.92 ± 1.33 **	96	0.98 ± 0.44	356	6.90 ± 0.63	Antagonism
FM55P	4.40 ± 0.43 *	72	1.51 ± 0.48	140	2.85 ± 0.61	Antagonism
FM55M2	5.53 ± 0.41	96	1.84 ± 0.59	132	5.29 ± 0.66	Additivity

The IC_50_ values (in µg/mL ± S.E.M.) for the mixture of CBD with CDDP were determined experimentally (IC_50 mix_) in four melanoma malignant cell lines in the in vitro MTT assay. The IC_50 add_ values were calculated from the lower and upper isoboles of additivity. The *n* _mix_—total number of items experimentally determined; *n* _add_—total number of items calculated for the additive mixture of CBD with CDDP; * *p* < 0.05 and ** *p* < 0.01 vs. the respective IC_50 add_ value.

**Table 3 ijms-23-06752-t003:** Isobolographic analysis of interactions between CBD and MTX (at the fixed ratio of 1:1) in malignant melanoma cell lines.

Cell Line	IC_50 mix_ (μg/mL)	n _mix_	Lower IC_50 add_ (μg/mL)	n _add_	Upper IC_50 add_ (μg/mL)	Interaction
SK-MEL 28	5.16 ± 0.57	96	1.59 ± 0.78	140	7.03 ± 1.00	Additivity
A375	6.57 ± 0.40	96	0.73 ± 0.25	306	6.79 ± 0.30	Additivity
FM55P	1.81 ± 0.33	72	1.04 ± 0.46	140	2.93 ± 0.49	Additivity
FM55M2	2.91 ± 0.28	96	2.22 ± 0.48	168	4.55 ± 0.64	Additivity

The IC_50_ values (in µg/mL ± S.E.M.) for the mixture of CBD with MTX were determined experimentally (IC_50 mix_) in four melanoma malignant cell lines in the in vitro MTT assay. The IC_50 add_ values were calculated from the lower and upper isoboles of additivity. The *n* _mix_—total number of items experimentally determined; *n* _add_—total number of items calculated for the additive mixture of CBD with MTX.

## Data Availability

Data are contained within the article.

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
