# Peer review of "Cannabidiol Interacts Antagonistically with Cisplatin and Additively with Mitoxantrone in Various Melanoma Cell Lines—An Isobolographic Analysis"

_ijms, 2022, doi:10.3390/ijms23126752_

Round 1
Reviewer 1 Report
Review of the manuscript which has been submitted to International Journal of Molecular Sciences-Manuscript no. ijms-1732235
Title: Cannabidiol interacts antagonistically with cisplatin and additively with mitoxantrone in various melanoma cell lines – an isobolographic analysis
In the current context of the study topic, the article entitled “Cannabidiol interacts antagonistically with cisplatin and additively with mitoxantrone in various melanoma cell lines – an isobolographic analysis” is very interesting, but the descriptions of the study motivation and the experimental part are deficient. From my point of view, the material sections must be written for everyone to understand, whether or not they are experts in the field. The introduction needs to be improved with additional documentation on the effects of CBD (as a suggestion: https://doi.org/10.3390/ijms23063344, https://doi.org/10.3390/ijms22189956, https://doi.org/10.3390/antiox10050738), as well as information about CDDP and MTX and where they are used. Also, what is the purpose of the paper, why were combinations of CBD, CDDP and MTX tested? Explain what is the isobolographic analysis and what is the purpose of it for the article (https://doi.org/10.3389/fphar.2019.01222 -this is a very clear article comparative with present material). Experimental methods are very few and incomplete, the results are very poorly explained and incomplete. Below I have made some punctual suggestions:
- Page 2, line 56; I do not think that further explanations are needed as “for more details see” to accompany the bibliography, the reference cited it enough.
- Figures: increase the resolution of the graphs, it is very hard to read them. Also, Figures 1,2,3 are tough to understand and to compare between the cell lines.
- Page 4, line 106; Figure 2 should be modified by increasing the resolution of the graphs. It is not clear in the format it is now. The individual graphs are small and no distinction can be made between cell lines. Because the control bar is at 100%, the other bars are very small and it is not clear what percentage they represent. For example, at first sight, there is no difference at all between HaCaT, which is the control, and cancer cell line (FM55M2, FM55P, etc.).
- Figure 3: why is the graph with HaCaT missing, which is the control cell line, so we can better understand the effect of the tested compounds and concentrations. My opinion is that the used concentrations are very high, considering that the pure compound was tested.
- Page 6, lines 125, 132; The resolution of the Supplementary Figures 1 and 2 are very low, the concentrations points are not visible. Please ask the authors to fix the problem.
- Page 6, lines 125, 132; I don't understand why the concentrations were expressed in logarithms (Supplementary Figures 1 and 2). What does it mean that the lines are not parallel? Please explain in detail so that those who are not specialists to understand;
- Page 6, lines 125, 132, 162, 195; For Supplementary Figures 1 and 2, Figure 4, 5, it is mandatory to include a chart with tests for the HaCaT cell line, otherwise, how can we know exactly that it does not kill normal cells along with malignant ones?
- Page 7, line 139; Please reformulate the sentence for a better understanding “With Student’s t-test with Welch correction” throughout the document, because it is not clear the senses of the experiments.
- Page 7, line 140; Please reformulate the sentence for a better understanding “With Student’s t-test with Welch correction the IC50 mix value significantly differed (p<0.05) from the IC50 add values (Table 2; t=2.013; df=196.1; p=0.045; Fig 4A).”
- Page 7, line 153; why do you perform “Isobolographic analysis of interactions between CBD and CDDP”.
- Page 12, line 344; some details concerning the isobolographic analysis must be added here as well, as the article does not make sense;
- Page 13, line 362; why did the authors do these tests if they claim that “To our best knowledge, the combination of CBD with CDDP should not be combined in melanoma patients due to the reduction of the anti-melanoma effects of the drugs in mixture.”
After making the indicated changes, the article may be suitable for publication.
Author Response
Replies to the Reviewers’ comments and suggestions
Title: Cannabidiol interacts antagonistically with cisplatin and additively with mitoxantrone in various melanoma cell lines – an isobolographic analysis
In the current context of the study topic, the article entitled “Cannabidiol interacts antagonistically with cisplatin and additively with mitoxantrone in various melanoma cell lines – an isobolographic analysis” is very interesting, but the descriptions of the study motivation and the experimental part are deficient. From my point of view, the material sections must be written for everyone to understand, whether or not they are experts in the field. The introduction needs to be improved with additional documentation on the effects of CBD (as a suggestion: https://doi.org/10.3390/ijms23063344, https://doi.org/10.3390/ijms22189956, https://doi.org/10.3390/antiox10050738), as well as information about CDDP and MTX and where they are used.
Reply: The introduction has been improved following the Reviewer’s suggestions and comments.
Also, what is the purpose of the paper, why were combinations of CBD, CDDP and MTX tested? Explain what is the isobolographic analysis and what is the purpose of it for the article (https://doi.org/10.3389/fphar.2019.01222 -this is a very clear article comparative with present material).
Reply: The purpose of this study has been clearly presented as suggested.
Experimental methods are very few and incomplete, the results are very poorly explained and incomplete.
Reply: Experimental methods based on isobolographic analysis have been described in more details. Results have been described more precisely, following both reviewers’ suggestions and comments.
Below I have made some punctual suggestions:
- Page 2, line 56; I do not think that further explanations are needed as “for more details see” to accompany the bibliography, the reference cited it enough.
- Reply: We have delated the unnecessary explanations, as suggested.
- Figures: increase the resolution of the graphs, it is very hard to read them. Also, Figures 1,2,3 are tough to understand and to compare between the cell lines.
- Reply: The resolution of the graphs has been increased to 1200 dpi, as suggested.
- Page 4, line 106; Figure 2 should be modified by increasing the resolution of the graphs. It is not clear in the format it is now. The individual graphs are small and no distinction can be made between cell lines. Because the control bar is at 100%, the other bars are very small and it is not clear what percentage they represent. For example, at first sight, there is no difference at all between HaCaT, which is the control, and cancer cell line (FM55M2, FM55P, etc.).
- Reply: The resolution of the graph has been increased to 1200 dpi, as suggested and the graph has been modified.
- Figure 3: why is the graph with HaCaT missing, which is the control cell line, so we can better understand the effect of the tested compounds and concentrations. My opinion is that the used concentrations are very high, considering that the pure compound was tested.
- Reply: In our opinion, there is no need to present proliferation of normal cells in the BrdU assay. Malignant melanoma cells proliferate within 72h due to their cancerous transformation, but normal cells should not proliferate. This is the reason not to test the studied drugs on normal human keratinocytes (HaCaT) line.
- Page 6, lines 125, 132; The resolution of the Supplementary Figures 1 and 2 are very low, the concentrations points are not visible. Please ask the authors to fix the problem.
- Reply: The resolution of both supplementary figures has been increased. We have used larger fonts and bolded points, as recommended. Additionally, following the Second Reviewer’s suggestion both graphs have been moved from Supplementary materials to normal figures.
- Page 6, lines 125, 132; I don't understand why the concentrations were expressed in logarithms (Supplementary Figures 1 and 2). What does it mean that the lines are not parallel? Please explain in detail so that those who are not specialists to understand;
- Reply: We have explained all these suggestions in the material and method section when describing in details the section on Isobolographic analysis as suggested by both reviewers.
- Page 6, lines 125, 132, 162, 195; For Supplementary Figures 1 and 2, Figure 4, 5, it is mandatory to include a chart with tests for the HaCaT cell line, otherwise, how can we know exactly that it does not kill normal cells along with malignant ones?
- Reply: The main goal of this paper was to find out the combination of two drugs that destroy melanoma cells, not normal cells.
- Page 7, line 139; Please reformulate the sentence for a better understanding “With Student’s t-test with Welch correction” throughout the document, because it is not clear the senses of the experiments.
- Reply: We have changed the mentioned sentences to better describe statistical analysis of data.
- Page 7, line 140; Please reformulate the sentence for a better understanding “With Student’s t-test with Welch correction the IC50 mix value significantly differed (p<0.05) from the IC50 add values (Table 2; t=2.013; df=196.1; p=0.045; Fig 4A).”
- Reply: We have changed the mentioned sentences to better describe statistical analysis of data
- Page 7, line 153; why do you perform “Isobolographic analysis of interactions between CBD and CDDP”.
- Reply: In in vitro studies, CDDP is used a reference drug, to compare the power and strength of the other drugs in eliminating the cancer cells. This is the reason to test interaction between CBD and CDDP to compare whether the combination of CBD with MTX was better than that of CBD with CDDP. In this study we confirmed evidently that the combination of CBD with MTX was better than that of CDDP with CBD.
- Page 12, line 344; some details concerning the isobolographic analysis must be added here as well, as the article does not make sense;
- Reply: The section on isobolographic analysis of interaction has been enlarged, as requested.
- Page 13, line 362; why did the authors do these tests if they claim that “To our best knowledge, the combination of CBD with CDDP should not be combined in melanoma patients due to the reduction of the anti-melanoma effects of the drugs in mixture.”
- Reply: After performing experiments we know now that the combination of CBD with CDDP should not be used clinically. We have changed the sentence to avoid ambiguity.
After making the indicated changes, the article may be suitable for publication.
Reviewer 2 Report
The authors presented interesting and valuable data, but the manuscript requires serious corrections.
1) Fig. 2 - the significance of differences should be taken into account.
2) Fig 3. The plot for the HaCaT line is missing.
3) The figures described as supplementary (1 and 2) are of very poor quality. They should be revised and incorporated into the main text. Why did the authors label the charts A, B, C, D, A ', B' ...? These markings do not appear in the text. These figures should be described in the text of the manuscript, at the moment their interpretation is difficult. The reader should not be guessing.
4) I think it would be worth doing additional research that could explain the results obtained by the Authors, e.g. inhibition of ERK phosphorylation.
Minor Notes:
Please add the structural formula of CBD and its chemical name;
Please format the tables as required by the journal.
Author Response
Replies to the Reviewers’ comments and suggestions
Comments and Suggestions for Authors
The authors presented interesting and valuable data, but the manuscript requires serious corrections.
1) Fig. 2 - the significance of differences should be taken into account.
Reply: Significance has been added to the Figure as suggested.
2) Fig 3. The plot for the HaCaT line is missing.
Reply: In our opinion, there is no need to present proliferation of normal cells in the BrdU assay. Malignant melanoma cells proliferate within 72h due to their cancerous transformation, but normal cells should not proliferate. This is the reason not to test the studied drugs on normal human keratinocytes (HaCaT) line.
3) The figures described as supplementary (1 and 2) are of very poor quality. They should be revised and incorporated into the main text. Why did the authors label the charts A, B, C, D, A ', B' ...? These markings do not appear in the text. These figures should be described in the text of the manuscript, at the moment their interpretation is difficult. The reader should not be guessing.
Reply: The resolution of both supplementary figures has been increased. We have used larger fonts and bolded points, as recommended. Additionally, both graphs have been moved from Supplementary materials to normal figures. Interpretation of these Figures is simple because they illustrate non-parallel concentrations-effect curves for the studied drugs when used alone.
4) I think it would be worth doing additional research that could explain the results obtained by the Authors, e.g. inhibition of ERK phosphorylation.
Reply: We plan to perform additional experiments to detect the mechanisms of action for the observed interaction between CBD and CDDP or MTX.
Minor Notes:
Please add the structural formula of CBD and its chemical name;
Reply: In our opinion, there is no need to present the formula and chemical name of cannabidiol. It is a well-known substance in research.
Please format the tables as required by the journal.
Reply: Tables have been formatted following the journal requirements.
Round 2
Reviewer 1 Report
Title: Cannabidiol interacts antagonistically with cisplatin and additively with mitoxantrone in various melanoma cell lines – an isobolographic analysis
In the current context of the study topic, the article entitled “Cannabidiol interacts antagonistically with cisplatin and additively with mitoxantrone in various melanoma cell lines – an isobolographic analysis” still has big flaws. The authors did not make the suggested changes even though they are relevant, instead, they repeatedly offered arguments that are not true.
The information about CDDP and MTX and where they are used is not added anywhere, such that “The introduction has” NOT “been improved following the Reviewer’s suggestions and comments”.
The purpose of the paper is described in several paragraphs, I have identified two (89-97; 389-396), it is confusing, it should be in one place and clearly delimited;
Paragraphs between lines 362-396; 405-416 should be included in the Introduction or Discussion sections. It's hard to believe that the corrections suggested by reviewers were not verified by a supervisor. The role of a reviewer is to follow other aspects, not to recommend arranging the literature information in an article.
The following affirmation is incorrect: “In our opinion, there is no need to present proliferation of normal cells in the BrdU assay. Malignant melanoma cells proliferate within 72h due to their cancerous transformation, but normal cells should not proliferate. This is the reason not to test the studied drugs on normal human keratinocytes (HaCaT) line.”. Whether it is cancerous or not, any cell proliferates over 72 hours. Furthermore, no matter what compounds you test, you need to have control of normal cells in addition to cancer cells. The cytotoxic effect of a drug or combination of drugs on malignant cells is not relevant without the results for cytotoxicity on normal cells. The basic problem with most chemotherapeutic agents is that they kill normal cells as well as malignant cells and cause side effects. Consequently, the experiments are irrelevant without a HaCaT graph.
In the Conclusions section, the authors repeat the same sentence.
Author Response
Title: Cannabidiol interacts antagonistically with cisplatin and additively with mitoxantrone in various melanoma cell lines – an isobolographic analysis
In the current context of the study topic, the article entitled “Cannabidiol interacts antagonistically with cisplatin and additively with mitoxantrone in various melanoma cell lines – an isobolographic analysis” still has big flaws. The authors did not make the suggested changes even though they are relevant, instead, they repeatedly offered arguments that are not true.
The information about CDDP and MTX and where they are used is not added anywhere, such that “The introduction has” NOT “been improved following the Reviewer’s suggestions and comments”.
Reply: Information on CDDP and MTX has been added as suggested.
The purpose of the paper is described in several paragraphs, I have identified two (89-97; 389-396), it is confusing, it should be in one place and clearly delimited;
Reply: The aim of this study has been presented in one place at the end of the Introduction, as suggested.
Paragraphs between lines 362-396; 405-416 should be included in the Introduction or Discussion sections. It's hard to believe that the corrections suggested by reviewers were not verified by a supervisor. The role of a reviewer is to follow other aspects, not to recommend arranging the literature information in an article.
Reply: Information on isobolographic analysis has been moved from the Material and method section to the Discussion, as suggested.
The following affirmation is incorrect: “In our opinion, there is no need to present proliferation of normal cells in the BrdU assay. Malignant melanoma cells proliferate within 72h due to their cancerous transformation, but normal cells should not proliferate. This is the reason not to test the studied drugs on normal human keratinocytes (HaCaT) line.”. Whether it is cancerous or not, any cell proliferates over 72 hours. Furthermore, no matter what compounds you test, you need to have control of normal cells in addition to cancer cells. The cytotoxic effect of a drug or combination of drugs on malignant cells is not relevant without the results for cytotoxicity on normal cells. The basic problem with most chemotherapeutic agents is that they kill normal cells as well as malignant cells and cause side effects. Consequently, the experiments are irrelevant without a HaCaT graph.
Reply: We have added the missing chart illustrating effects of cannabidiol on HaCaT line in the BrdU test as suggested.
In the Conclusions section, the authors repeat the same sentence.
Reply: We have deleted the similar sentence from the Conclusions.
Reviewer 2 Report
The manuscript has been greatly improved and, in my opinion, is suitable for publication. I congratulate the Authors on their good job.
Author Response
Thank you very much for your valuable comments and suggestions
Round 3
Reviewer 1 Report
Review of the manuscript which has been submitted to International Journal of Molecular Sciences-Manuscript no. ijms-1732235
Title: Cannabidiol interacts antagonistically with cisplatin and additively with mitoxantrone in various melanoma cell lines – an isobolographic analysis
In the current context of the study topic, the article entitled “Cannabidiol interacts antagonistically with cisplatin and additively with mitoxantrone in various melanoma cell lines – an isobolographic analysis” is very interesting.
After sustained efforts, the Introduction is cursive, presenting the scientific context and research purpose of the study.
The Results section is clearer; adding the graph of CBD cytotoxicity to normal keratinocytes reinforces the negative effects of CBD on malignant cell lines. At the same time, the Discussion section complements and scientifically argues the results obtained.
In this presentation form, the article is suitable for publication in International Journal of Molecular Sciences.